# Multi-Level Control of the ATM/ATR-CHK1 Axis by the Transcription Factor E4F1 in Triple-Negative Breast Cancer

**DOI:** 10.3390/ijms23169217

**Published:** 2022-08-16

**Authors:** Kalil Batnini, Thibault Houles, Olivier Kirsh, Stanislas Du Manoir, Mehdi Zaroual, Hélène Delpech, Chloé Fallet, Matthieu Lacroix, Laurent Le Cam, Charles Theillet, Claude Sardet, Geneviève Rodier

**Affiliations:** 1Institut de Recherche en Cancérologie de Montpellier, IRCM U1194, Montpellier University, INSERM, ICM, CNRS, 34090 Montpellier, France; 2Epigenetics and Cell Fate Team, CNRS, University Paris Cité, 75006 Paris, France

**Keywords:** E4F1 transcription factor, DNA Damage Response (DDR), ATM/ATR-CHK checkpoint pathway, triple-negative breast cancer, chemotherapy

## Abstract

*E4F1* is essential for early embryonic mouse development and for controlling the balance between proliferation and survival of actively dividing cells. We previously reported that E4F1 is essential for the survival of murine p53-deficient cancer cells by controlling the expression of genes involved in mitochondria functions and metabolism, and in cell-cycle checkpoints, including *CHEK1*, a major component of the DNA damage and replication stress responses. Here, combining ChIP-Seq and RNA-Seq approaches, we identified the transcriptional program directly controlled by E4F1 in Human Triple-Negative Breast Cancer cells (TNBC). E4F1 binds and regulates a limited list of direct target genes (57 genes) in these cells, including the human *CHEK1* gene and, surprisingly, also two other genes encoding post-transcriptional regulators of the ATM/ATR-CHK1 axis, namely, the TTT complex component TTI2 and the phosphatase PPP5C, that are essential for the folding and stability, and the signaling of ATM/ATR kinases, respectively. Importantly, E4F1 also binds the promoter of these genes in vivo in Primary Derived Xenograft (PDX) of human TNBC. Consequently, the protein levels and signaling of CHK1 but also of ATM/ATR kinases are strongly downregulated in E4F1-depleted TNBC cells resulting in a deficiency of the DNA damage and replicative stress response in these cells. The E4F1-depleted cells fail to arrest into S-phase upon treatment with the replication-stalling agent Gemcitabine, and are highly sensitized to this drug, as well as to other DNA-damaging agents, such as Cisplatin. Altogether, our data indicate that in breast cancer cells the ATM/ATR-CHK1 signaling pathway and DNA damage-stress response are tightly controlled at the transcriptional and post-transcriptional level by E4F1.

## 1. Introduction

Initially identified as a cellular target of the E1A viral oncoprotein [1,2], the zinc-finger transcription factor E4F1 was then described as a physical or genetic interactor of several tumor suppressors and oncogenes that gate cell division, survival and DNA repair in proliferating cells, including pRB, RASSF1A, p14^ARF^, BMI1, FHL2, CHK1, p53, PARP-1 and BRG1 [3,4,5,6,7,8,9,10,11]. E4F1 was also shown to act as an atypical ubiquitin E3-ligase on p53 that does not promotes p53 degradation, but rather modulates its transcriptional activities, directing the p53-response towards the expression of the genes involved in cell-cycle arrest rather than cell death [8]. As shown in *E4f1*-KO mice, E4F1 is also essential for early embryonic mouse development [12] and tissue homeostasis [13,14], and for either proliferation or survival of actively dividing cells, including fully transformed cells [10,11,15,16,17]. To elucidate the transcriptional program controlled by E4F1 in murine proliferating cells, we previously performed genome-wide omics analyses to identify the genes that were both directly bound and regulated by E4F1 in the normal and transformed Mouse Embryonic Fibroblasts (MEFs), and in Embryonic Stem (ES) cells (combining chromatin immunoprecipitation followed by deep sequencing (ChIP-Seq) and transcriptomic analyses of differentially expressed mRNAs between E4F1 WT and KO cells). These analyses indicated that E4F1 binds and directly controls a small number of target genes (between 50 and 100 genes depending on the situation and cell type), including the genes implicated in the cell-cycle checkpoints and survival, but also unexpectedly, in a transcriptional program (about 20/30 genes) involved in mitochondria functions and metabolism [17,18]. In particular, E4F1 coordinates the transcription of genes coding for several components or regulators of the PDH-mediated pyruvate oxidation pathway, with marked metabolic consequences in E4F1 KO murine cells and tissues [19,20,21]. Noteworthy, this profiling of E4F1 target genes in the mouse also showed that E4F1 directly binds and regulates the *Chek1* gene encoding the Checkpoint kinase 1 (CHK1) [17,18,22], a central player of the DNA replication stress and damage responses. This connection of E4F1 with the DNA stress response was further supported by recent reports showing that the E4F1 protein also directly impacts on the CHK1 protein stability [10] and is recruited to DNA breaks where it promotes the ATR-CHK1 signaling and, together with PARP1, the recruitment of BRG1 to DNA lesions [11]. In the present study, we reveal yet another new level of control of E4F1 on this DNA stress-response signaling pathway in Human Triple-Negative Breast Cancer cells (TNBC) through the upstream regulators of the ATM/ATR proteins. 

The ataxia telangiectasia-mutated serine/threonine kinase (ATM)/checkpoint kinase 2 (CHK2) and the ATM and Rad3-related serine/threonine kinase (ATR)/checkpoint kinase 1 (CHK1) cascades are the two major signaling pathways driving the response to the DNA damage (DDR), and in the case of ATR-CHK1, also to the replicative stress induced by replication fork-stalling. By phosphorylating the key gate keepers of proliferation and cell death, such as p53, cdc25 phosphatases or histone H2AX, they signal and prevent the cells bearing DNA damage or the stalled replication forks from progressing through the cell cycle [23,24]. This pathway is essential for the preservation of genomic stability and acts as a barrier against tumorigenesis and cancer progression. Moreover, since the activation of the oncogenes in cancer cells often leads to DNA replication stress, the cancer cells heavily rely on this replication checkpoint machinery for survival. Hence, the inactivation of the ATM-CHK2 and ATR-CHK1 pathways efficiently sensitizes the malignant cells, in particular p53-deficient cancer cells, to DNA damage or high levels of replication stress induced by radiotherapy or chemotherapies. Numerous, promising preclinical and clinical studies suggest that ATM, ATR, CHK1, CHK2 and WEE1 inhibitors or modulators, alone or in combination with other therapeutics, could be potent therapeutic options for certain cancers, including TNBC [24,25,26,27,28]. Understanding the molecular and cellular regulatory mechanisms of the ATM/ATR-CHK1/CHK2 axis in human cancer cells is then essential. Hence, these two kinases cascades are tightly regulated at multiple levels, including by controlling the expression of their genes and the location, folding, stability, phosphorylation and interactions of the proteins that compose them.

Here we identified the transcriptional program directly controlled by E4F1 in Human triple-negative breast cancer (TNBC) cells. This analysis points in particular at the central role of E4F1 in the control of the ATM/ATR-CHK1 cascade, through its control of the human *CHEK1* gene, but also of *PPP5C* and *TTI2*, two of the genes that are involved in the signaling and in the protein folding and stability of the ATM/ATR kinases. The depletion of E4F1 in the TNBC cells results in the downregulation of the ATM/ATR-CHK1 cascade that sensitizes these cells to chemotherapy (CT) drugs.

## 2. Results and Discussion

The initial observation that prompted us to study E4F1 in Human Triple-Negative Breast Cancer (TNBC) cells was that its shRNA-mediated depletion in two TNBC cell lines, SUM159 (Figure 1) and HCC38 (Appendix A), strongly sensitized these cells to the chemotherapy (CT) drugs, Gemcitabine and Cisplatin. Hence, the monitoring of the cell cycle profiles of SUM159 (Figure 1B,C) and HCC38 cells (Appendix A) by flow cytometry, showed that the E4F1-depleted cells failed to arrest in the S-Phase upon treatment with the replication poison, Gemcitabine, when compared to the control cells. At the sub-lethal dose of these drugs (IC30, as determined in the lab for SUM159 and HCC38 cells), the E4F1-depletion was associated with a strong increase in cell death, as assessed by the change of cellular morphology (phase contrast microscopy; Figure 1D) or by flow cytometry analysis of annexin-V/PI co-labelling (Figure 1E,F/SUM159 and Appendix A). To address the potential off-target effects of the E4F1 shRNA, rescue experiments were performed in the SUM159 (Figure 1E,F) and HCC38 (Appendix A) cells by co-transducing cells with an expression vector encoding a RNAi-resistant human E4F1 cDNA (pE4F1*). In both of the cell lines, the sensitization of the E4F1-depleted cells towards the CT drugs was significantly rescued by this construct. Altogether, these results indicated that the E4F1-depleted human TNBC cells had an abnormal response to both, a DNA damaging drug that produces DNA cross-links and adducts (Cisplatin), and to a DNA replication poison (the nucleoside analogue, Gemcitabine).

This sensitization to the DNA damage and replication stress, as well as the failure to arrest in presence of a replication poison suggested that E4F1 could control the expression of factors involved in the checkpoints that allow the cells to stop when they are subjected to these insults [24,25,26,27,28]. To identify such E4F1 targets, we next determined by genome-wide analyses the transcriptional program directly controlled by E4F1 in the SUM159 cells (i.e., the genes whose promoter regions were bound by E4F1 and found differentially expressed in E4F1-depleted cells) by combining E4F1 chromatin immunoprecipitation—followed by deep sequencing (ChIP-Seq) analyses and RNA-Seq analyses of E4F1-depleted (E4F1-shRNA-mediated). On one hand, these RNA-seq analyses (independent experiments performed on the sh*E4F1*- or sh*Ctrl*-treated cells) revealed that the mRNA levels of a large set of genes (>8000, *p* < 0.05) were deregulated, but for the vast majority of them, the changes were rather modest in the E4F1-depleted cells compared to the control (GSE128099, GSE128159) (Figure 2A). On the other hand, ChIP-Seq analyses (independent experiments performed with a polyclonal anti-E4F1 antibody, or in control conditions without primary antibody, as previously described on mouse cells [17,18]), revealed that only 116 human genome regions were significantly and reproducibly bound by E4F1 (GSE128104, GSE128159) (Appendix A), 102 of them being located in the ±3 kb regions around a transcription start site (TSS) of a gene (Figure 2B; Appendix A), a majority of them even locating in the −200/+200 bp of TSS), and 94 of them in regions defined as promoters (Appendix A). The MEME logo sequence analysis of these E4F1-binding sites in the SUM159 cells revealed a human E4F1 consensus motif at the center of peak regions that is very similar to the mouse E4F1 consensus motif (Figure 2C) previously identified by our laboratory in Mouse embryo fibroblasts (MEF) and mouse embryonic stem cells (ES) [17,18,19,20]. Moreover, a significant overlap exists between this list of 102 human genes (±3kb regions) bound by E4F1 in TNBC cells and the equivalent list of 109 genes bound by E4F1 in mouse embryonic cells (GSE57228, [17,18]). We recalled this list compiling the E4F1 ChIP-seq peaks identified in MEFs and ES cells in Appendix A. About half of the E4F1-target genes we detected in the human TNBC cells were also bound by E4F1 in the mouse cells of different lineages (fibroblasts and ES cells), suggesting a significant conservation of the transcriptional program directly controlled by E4F1 in the different cell types and different species (Venn diagram; Appendix A. The 58 human genes bound by E4F1 in TNBC that are also bound by E4F1 in mouse cells are listed in Appendix A).

The overlap between the ChIP-Seq and RNA-seq datasets, generated in the SUM159 cells, identified a short list of 57 genes directly bound and regulated by E4F1 in these cells (Figure 2D,E). As previously observed in the murine cells [17], a majority of these human genes were downregulated in E4F1-depleted SUM159 cells, suggesting that E4F1 acts mainly as a transactivator for its target genes. As in the mouse cells [17], the *E4F1* gene itself is a direct target of the E4F1 transcription factor in human cells. 

A first survey of this short list of E4F1-target genes in human TNBC cells, confirmed that as previously observed in mouse cells and tissues [17,18,19,20], E4F1 controls a set of genes important for mitochondrial homeostasis and metabolic functions, including, *DLAT*, *NUDT9*, *TAZ*, *CRLS1*, *PDSS2*, *TOMM7*, *DMAC1* and *MRPS18C*.

A closer study of this list in search for the E4F1 target genes that could be involved in DNA damage-response and replication stress checkpoint in TNBC, revealed that at least 6 genes out of the 57 encode key players of these processes, including: (i) FAAP100, a poorly described but essential component of the Fanconi Anemia (FA) core complex that regulates FANCD2 mono-ubiquitination [29,30]. The FA complex plays a central role in the DNA damage-response network, involving the breast cancer susceptibility gene products, BRCA1 and BRCA2; (ii) SFR1, a component of the trimeric RAD51-SWI5-SFR1 complex involved in homology-directed DNA repair [31,32], which also interacts with the estrogen receptor alpha (ERα) in breast cancer cells [33]; (iii) PARG, the main Poly(ADP-ribose) glycohydrolase which endo- and exoglycosidase activity can reverse the action of PARP enzymes, and therefore thought to be involved in the control of the repair of the DNA strand breaks and for continued replication at the perturbed replication forks [34]. Interestingly, PARG also interacts with the replication factor PCNA and is strongly recruited at DNA damage sites [35] as recently proposed for E4F1 which is recruited to DNA lesions where, together with PARP1, it stimulates the recruitment of BRG1 [11]. Outside of the scope of this manuscript, the future studies on this intriguing connection between PARG, PARP1, E4F1 and BRG1, might reveal a novel mechanism of DNA repair with complex feedback control loops; (iv) As previously reported in the mouse embryonic cells [17] E4F1 also clearly binds and controls the expression of the human *CHEK1* gene in TNBC cells (Figure 2E and Figure 3). *CHEK1* encodes the serine threonine kinase CHK1, a key mediator of the DNA damage and replication stress response that prevents the damaged and stressed cells from progressing through the cell cycle [24,25,26,27,28]. CHK1 is activated by the ATR kinase and lies at the heart of the signaling cascades activated in response to both DNA damage and replication stress (ATR-CHK1 pathway). These signaling pathways are now considered as therapeutic targets to boost the efficiency of chemotherapies in cancer, including in triple-negative breast cancer (TNBC) [22,24,25,26,27,28]; (v) Finally, E4F1 also binds and controls the *PPP5C* and the *TTI2* genes, coding, respectively, for the PPP5C phosphatase and for a subunit of the TTT complex that both act as post-transcriptional regulators of the ATM/ATR kinases. Indeed, following the exposure to ionizing radiation or UV light, PPP5C has been reported to contribute to ATM-, ATR- and CHK1-mediated checkpoint regulation. Consistently, *ppp5c*^−/−^ MEFs exhibited increased sensitivity to various DNA-damaging agents and replication stress inducers [36,37,38]. TTI2 is an essential subunit of the molecular co-chaperone, termed the TELO2-TTI1-TTI2 (TTT) complex, involved in the folding, maturation and likely stability of the Phosphatidylinositol 3-kinase-related kinases (PIKKs) family, including ATM and ATR [39,40]. This specialized complex recognizes the newly synthesized ATM and ATR kinases, and connects them with chaperones including, the Hsp90 chaperone. Targeting any component of the TTT complex results in a rapid degradation and depletion of ATM/ATR [41,42,43], hence making this co-chaperone a key player in the cellular resistance to DNA damage stresses [39]. Figure 2E4F1 directly controls a limited set of genes in the TNBC cell line SUM159, including a sub-program involved in the ATM/ATR-CHEK signaling pathway. (**A**)—Fold change plot of differentially expressed genes, assessed by RNA-Seq, in SUM159 cells four days after the initial exposure to either sh*Ctrl* or sh*E4F1*. The expression of a large set of genes is moderately deregulated in E4F1-depleted cells compared to control; (**B**)—Identification of E4F1 target genes by ChIP-Seq. E4F1 ChIP-Seq read densities in the 3 kb regions surrounding the Transcription Start Sites (TSS) of the closest genes located nearby E4F1 binding sites, as identified in SUM159 cells. Out of the 116 ChIP-Seq peaks identified by ChIP-Seq in SUM159 cells (*n* = 2; Appendix A, 102 were in these 3kb regions and 94 in promoters, indicating that E4F1 is mainly located nearby or on the TSS of its target genes; (**C**)—MEME logo sequence analysis of the DNA fragments bound by E4F1 in SUM159 cells reveals a human E4F1 consensus motif that is similar to the Mouse E4F1 consensus motif previously identified by our laboratory [17]; (**D**)—Venn diagram of the overlap between all of the E4F1 bound genes identified by ChIP-Seq in SUM159 cells (GSE128104 (in red)) and genes differentially expressed (*p* < 0.05) between SUM159 cells treated with either sh*Ctrl* or sh*E4F1* (GSE128099 (in blue)). These genome-wide analyses were performed at a time where all of the cells treated with shRNAs were still viable (annexin-V negative) and still actively growing. The overlap defines a set of 57 genes directly bound and regulated by E4F1 in SUM159; (**E**)—List of the 57 E4F1 direct target genes as defined in panel D. Table provides information about the differential expression of transcripts in SUM159 cells treated with either sh*Ctrl* or sh*E4F1* (Ratio KO/WT), the distance of the E4F1 consensus motif to the transcription start site (TSS) of the gene, and whether these genes code for factors involved in cellular stress response and checkpoints (red+) including proteins involved in the ATM/ATR-CHK kinases axis (CHK1, PPP5C and TTI2).
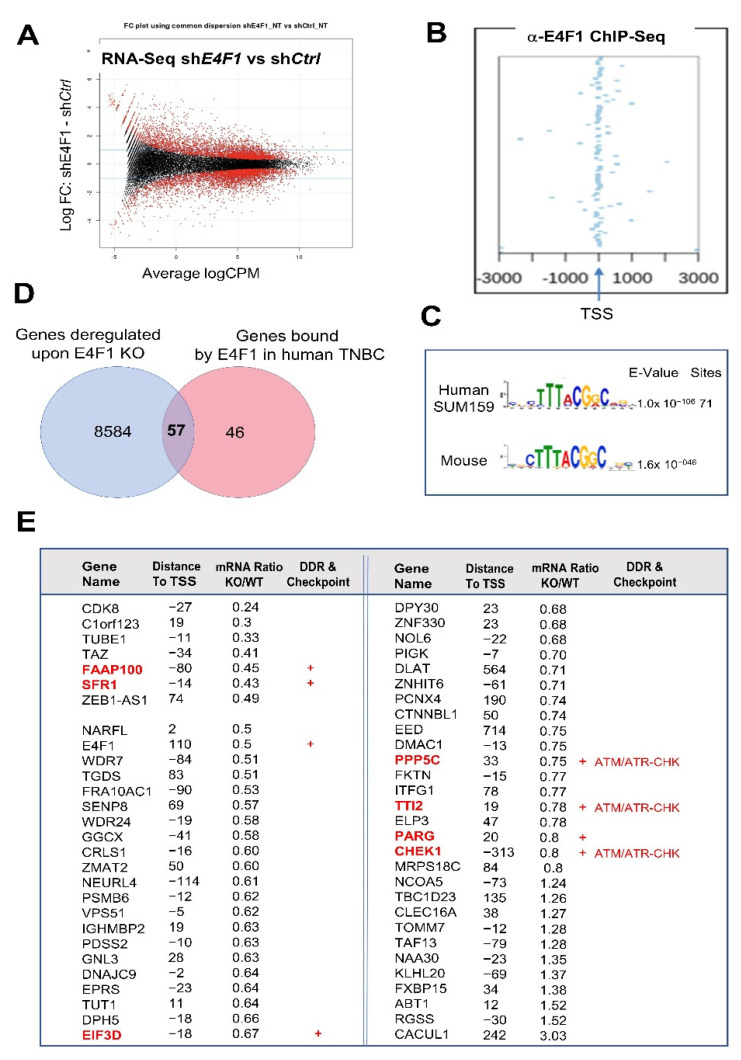



Based on this co-regulation of *CHEK1*, *PPP5C* and *TTI2*, we hypothesized that these three genes might define an E4F1-dependent and *hitherto* undescribed ATM/ATR-CHK1 “regulon”, which combined downregulation in the E4F1-depleted TNBC cells, could result in a functional inactivation of both the CHK1 and ATM/ATR activities. As this could provide a molecular explanation for the high sensitivity of the E4F1-depleted TNBC cells to CT drugs, we further investigated this sub-group of E4F1 targets. To validate these three genes as direct targets of E4F1, we designed the qPCR primers according to the E4F1 ChIP-seq peak profiles at their start site (TSS) regions (Figure 3A) and undertook ChIP-qPCR validation experiments in control and the E4F1-depleted SUM159 cells. These assays confirmed the presence of the E4F1 protein on the TSS region of *CHEK1*, *PPP5C*, *TTI2* and *E4F1* genes in the control cells, and showed a significant decrease in this recruitment in the E4F1-depleted cells (Figure 3B). In contrast, no E4F1 binding was detected at the *CHK2* promoter region or at an intergenic region (NC3), used as controls in these ChIP-qPCR experiments (Figure 3B). To assess whether E4F1 also binds this regulon in TNBC tumors in vivo, the ChIP-qPCR analyses were also performed on freshly isolated samples prepared from primary Human TNBC tumor-derived xenografts (PDX) [44]. Importantly, the qPCR primers used in this experiment were specific of the human genes, making these ChIP-qPCR analyses specific to human tumor cells and not to murine stroma present in the PDX tumor. As shown in Figure 3C, a recruitment of the E4F1 protein at the promoter regions of the *CHEK1*, *PPP5C*, *TTI2* and *E4F1* genes was also observed in this human tumor model, indicating that the control of these genes by E4F1 in TNBC extends beyond the cells in culture. Finally, the RT-qPCR validation experiments confirmed that the mRNA levels of these genes were downregulated upon E4F1-depletion in the SUM159 cells (Figure 3D; Appendix A). Figure 3E4F1 directly binds and controls the *CHEK1*, *PPP5C*, *TTI2* and *E4F1* genes in the TNBC cell line SUM159 and in a primary TNBC tumor-derived xenograft (TNBC PDX). (**A**)—ChIP-Seq read densities detected at the *CHEK1*, *PPP5C*, *TTI2* genes and at the *E4F1* gene itself, in SUM159 cells using either anti-E4F1 [17] or control antibodies. Tag densities were calculated using the HOMER Suite and visualized with CISGENOME browser; (**B**)—ChIP-qPCR validation of the presence of E4F1 on the promoter region of *CHEK1*, *PPP5C*, *TTI2* and *E4F1* genes in SUM159 cells, 5 days after their infection with lentiviral vectors expressing shRNAs directed against either E4F1 (sh*E4F1*) or control sequence (sh*Ctrl*). The *CHEK2* promoter region and an intergenic region (NC3) are used as controls. Enrichments are represented as percentage of input (data are means ± SD, *n =* 3 independent experiments); (**C**)—ChIP-qPCR analysis of the presence of E4F1 protein on the promoter region of *CHEK1*, *PPP5C*, *TTI2* and *E4F1* genes in primary Human Triple-Negative Breast tumor-derived xenograft [44]. ChIP-qPCR analyses were performed on freshly isolated tumor samples, two weeks after their engraftment on *Nude* mice. Enrichments are represented as percentage of input (data are means ± SD, *n =* 3 distinct tumors); (**D**)—mRNA levels of E4F1 target genes coding for CHEK1, PPP5C and TTI2 in SUM159 cells treated with either shCtrl or sh*E4F1*. Measured by RT-qPCR, 5 days after cell infection with lentiviral vectors expressing shRNAs. Data are means ± SD, *n =* 3 independent experiments. NS: non significative, ** *p* < 0.01, *** *p* < 0.001 as determined by a one way ANOVA test performed on GraphPad Prism.
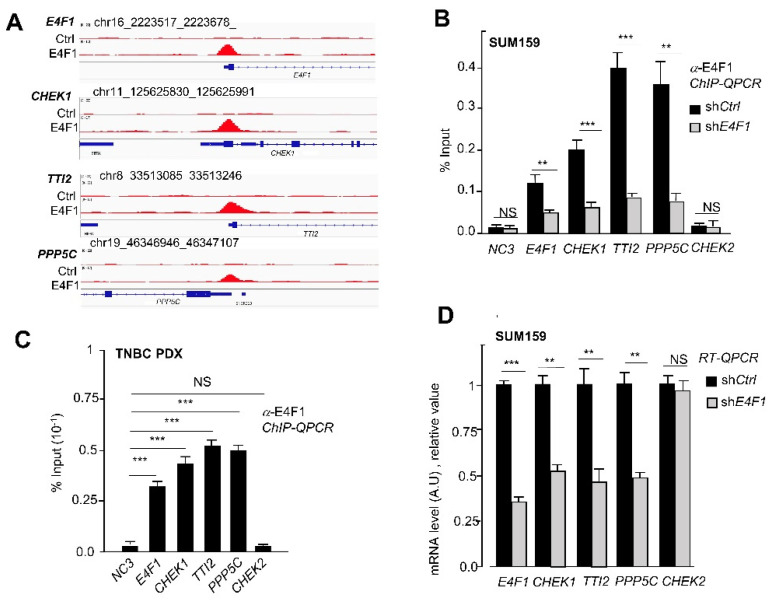



As a consequence of the downregulation of the mRNA levels, the CHK1, TTI2 and PPP5C protein levels were also markedly decreased in both the E4F1-depleted SUM159 (Figure 4A) and HCC38 (Appendix A) TNBC cells, as shown by immunoblotting. To confirm these downregulations were not due to an off target effect of the shRNA, we used a second shRNA targeting E4F1 (sh*E4F1*#2) in SUM159 and obtained similar results on the levels of the ATM, ATR, CHK1, PPP5C and TTI2 proteins (Appendix A). We next assessed whether the low levels of TTI2 in the E4F1-depleted TNBC cells also indirectly impacted the ATM and ATR protein levels, as reported in other cellular models [39,40,41,42,43]. We first verified that TTI2 depletion actually resulted in a downregulation of ATR and ATM protein in TNBC cells. The siRNA-mediated depletion of TTI2 was obtained in SUM159 cells and indeed triggered a disappearance of the ATR and to a lesser extend ATM proteins in these cells (Figure 4C). Hence, low levels of TTI2 in *E4F1*-depleted cells was consequently associated with a parallel marked loss of ATR protein in both SUM159 (Figure 4A and Figure 5B,C) and HCC38 cell lines (Appendix A). We could also observe a loss of ATM protein in the sh*E4F1*-treated SUM159 (Figure 4A and Figure 5B,C), but could not confirm this in HCC38, as we failed to detect the ATM protein in this cell line. Importantly, normal TTI2, as well as ATR and ATM protein levels were restored by co-transducing E4F1 shRNA-treated cells with the expression vector encoding a RNAi-resistant human E4F1 cDNA (pE4F1*) (Figure 5A; Appendix A). A further important finding was that *ATM* and *ATR* mRNA levels, measured in SUM159 cells, were unaffected by *E4F1* depletion (Figure 4B), excluding a scenario in which E4F1 would have directly or indirectly regulated the transcription of these genes. Considering that the alterations of the TTT complex results in PIKKKs (including ATM and ATR) misfolding, and that such misfolding might result in their degradation in lysosomes [39,45], we reasoned that the ATR/ATM protein level could be restored in the E4F1-depleted cells by exposing the cells to a lysosome inhibitor. Hence, the treatment of the E4F1-depleted SUM159 cells with the lysosomal inhibitor E64 resulted in a partial restoration of the ATR protein level (Figure 4D), further supporting the notion that the control of E4F1 on ATR expression actually occurred at the post-transcriptional level.

Altogether, these results indicate that the transcriptional regulation of *TTI2* by E4F1 leads to an indirect control of E4F1 on the ATM and ATR protein levels in the TNBC cells. Figure 4The transcriptional control exercised by E4F1 on the TTT complex component TTI2, strongly impacts on ATM and ATR protein levels in the TNBC cell line, SUM159. (**A**)—E4F1 depletion impacts on TTI2, ATM and ATR protein levels. Immunoblot analysis of indicated proteins in total extracts prepared from SUM159 cells treated with either sh*Ctrl* or sh*E4F1*. Extracts were prepared 4 days after cell infection with lentiviral vectors expressing shRNAs; (**B**)—E4F1 depletion impacts on TTI2 mRNA level, but not on ATM and ATR mRNA levels, nor on the mRNA level of TTI1, another component of the TTT complex. Measured by RT-qPCR, 5 days after cell infection with lentiviral vectors expressing either sh*Ctrl* or sh*E4F1*. Data are means ± SD, *n*= 3 independent experiments. ** *p* < 0.01, as determined by a one way ANOVA test performed on GraphPad Prism; (**C**)—siRNA-mediated depletion of TTI2 impacts on ATR and ATM protein levels in SUM159. Immunoblot analysis of indicated proteins in total extracts prepared from SUM159 cells treated for 48 h with either a control siRNA (siCtrl) or an siRNA directed against human TTI2 mRNA; (**D**)—Immunoblot analysis of indicated proteins in total extracts prepared from SUM159 cells treated with either sh*Ctrl* or sh*E4F1* and with the lysosomal inhibitor E64 (10 μM).
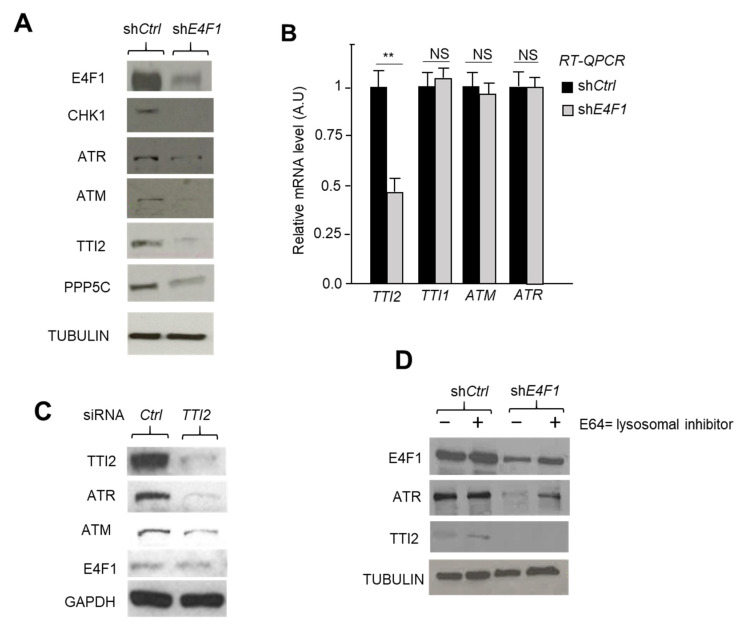



To confirm that the ATM/ATR-CHK1 signaling pathway is functionally downregulated in the E4F1-depleted cells, we next exposed the control or E4F1-shRNA-transduced cells to the DNA damaging agents, Cisplatin or Gemcitabine, and then monitored the phosphorylation levels of γH2AX and CHK1 in these cells, two well-established biomarkers of the ATM/ATR activity in response to DNA damage [22,46]. Consistently, the immunoblotting analyses with phospho-specific antibodies showed that *E4F1* depletion in SUM159 or HCC38 cells impacted on both CHK1 protein level and on its Serine 345 phosphorylation by ATM/ATR in response to Gemcitabine and Cisplatin. This effect was rescued upon the co-expression of the shRNA resistant pE4F1* cDNA (Figure 5A). Likewise, in the same conditions, the immunofluorescence and immunoblotting analyses also revealed that the ATM/ATR-mediated phosphorylation of γH2AX on serine 139 was markedly decreased in the *E4F1*-depleted cells compared to the control cells upon treatment with both of the CT drugs (Figure 5B,C; Appendix A). Of note, the *E4F1* depletion did not impact on the formation of 53BP1 foci in response to Gemcitabine. The preservation of this marker of the early phase of the DNA damage-response, suggest that the early steps of this response, upstream of the ATM/ATR-CHK1 signaling, might still be functional in the E4F1-depleted cells (Figure 6) [47].Figure 5E4F1-depletion impacts on ATM/ATR-CHK1 signaling in response to CT drugs Gemcitabine and Cisplatin. (**A**)—ShRNA-mediated depletion of E4F1 in SUM159 TNBC cells impacts on CHK1 protein level and on the S^345^ phosphorylation of CHK1 by ATM/ATR in response to Gemcitabine and Cisplatin. This effect was reversed by the co-expression of the shRNA resistant pE4F1* cDNA. Immunoblot analysis of indicated proteins or phosphor-proteins (CHK1^P^) in total extracts prepared from SUM159 cells transduced with either control shRNA (sh*Ctrl*), sh*E4F1*, or sh*E4F1* + pE4F1*, and treated for 24 h with Gemcitabine or Cisplatin. See also immunoblots in Appendix A showing a similar experiment in another TNBC cell line, HCC38; (**B**,**C**) Immunofluorescence staining of DNA (blue), phospho (S139)- γH2AX (green) and ATR (red) in SUM159 cells transduced with either sh*Ctrl*, sh*E4F1*, or sh*E4F1* + pE4F1*, and upon antibiotics selection, treated for 24 h with Gemcitabine; (**B**) Representative experiment; (**C**) Quantitation of three independent experiments expressed as mean values ± SD. *** *p* < 0.001 as determined by two-tailed unpaired *t*-test performed on Graph Pad Prism. See also Appendix A for immunoblotting analysis of γH2AX phosphorylation in cell lysates.
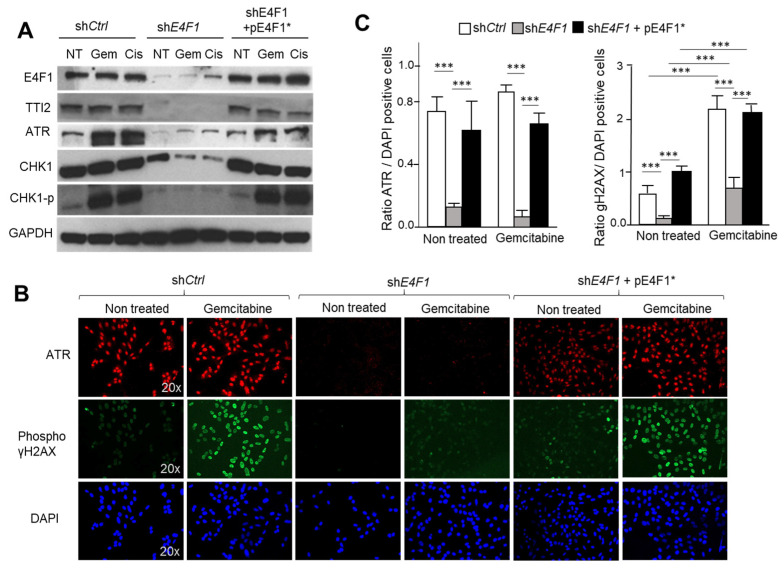

Figure 6E4F1-depletion does not impact on the formation of 53BP1 foci in response to Gemcitabine, suggesting the early step of the DNA damage-response, upstream of the ATM/ATR-CHK1 signaling, is still functional in E4F1-depleted cells. A and B- Immunofluorescence staining of DNA (blue) and of 53BP1 protein (red) in SUM159 cells transduced with either control sh*Ctrl* or sh*E4F1* and, upon antibiotics selection, treated for 24 h with Gemcitabine. (**A**) representative experiment; (**B**) Quantitation of independent experiments expressed as mean values ± SD of three independent experiments. *** *p* < 0.001 as determined by two-tailed unpaired *t*-test performed on Graph Pad Prism.
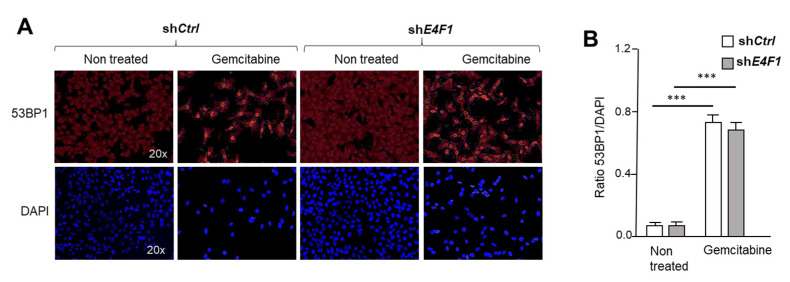



As summarized in Figure 7, this work, together with the previous reports in murine cells [10,11,17] reveal that the ATM/ATR–CHK1 signaling pathway is tightly controlled at the transcriptional and post-transcriptional levels by E4F1. In brief, E4F1 exerts multilevel controls on this stress checkpoint; it binds at the start site and controls the expression of the *CHEK1* gene (as observed in the mouse embryonic cells by [17,22], and here in human TNBC). It also directly interacts with the CHK1 Kinase, a protein–protein association that modulates CHK1 protein stability and activity [10] and together with PARP-1 acts to recruit BRG1 to DNA lesions, where it cooperates in the ATM/ATR DNA repair signaling [11]. Figure 7Schematic representation of the different levels of controls exercised by E4F1 on mitochondrial metabolism and the ATM/ATR–CHK1 axis. E4F1 is an ubiquitously expressed transcription factor that binds a limited set of genes (around 100), including its own promoter, as reported in mouse [17,18] and here in human cells. On the one hand, E4F1 controls genes important for mitochondrial homeostasis and metabolic functions, including a set of genes involved in pyruvate oxidation [19,20], that we termed a “PDH regulon” and which include *DLAT, DLD, MPC1* and *slc25A1*). On the other hand, E4F1 directly binds and controls the expression of the *CHEK1* gene, as observed in mouse cells [17] and here in human TNBC cells. Here, we also found that E4F1 directly binds and regulates two other genes, *TTI2* and *PPP5C,* encoding upstream post-translational regulators of the ATM/ATR-CHK1 signaling pathway. TTI2 is a component of the molecular co-chaperone TTT complex which specifically recognizes the newly synthesized ATM and ATR kinases, and is essential for their correct folding and stability [39,41]. PPP5C is a protein phosphatase that activates ATM, ATR and CHK1 kinase activities [36,37,38]. Other recent reports show that the E4F1 protein also directly interacts with the CHK1 Kinase to modulate CHK1 protein stability and activity [10], and also cooperates with PARP-1 and BRG1 to trigger ATM/ATR signaling at DNA lesions [11]. Altogether, this suggests the existence of a hitherto unidentified “ATM/ATR-CHK1 regulon” controlled by E4F1.
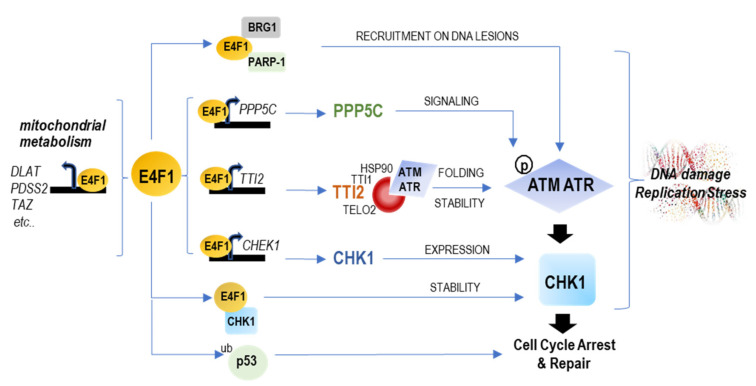



In addition, we now reveal in this report that E4F1 also directly controls two further genes in the Human TNBC cells, *TTI2* and *PPP5C,* that encode key post-transcriptional regulators of the ATM/ATR protein folding/stability and signaling, respectively, with consequences on the ATM/ATR proteins’ level. Although not addressed in this report, the control of the expression of the phosphatase PPP5C by E4F1 might add yet another level of important control of the ATM/ATR signaling and activities by E4F1 that remains to be investigated. Another limitation of the present study is that we cannot rule out at this stage that the other genes identified here as E4F1 targets could also be involved in the observed sensitization of E4F1-depleted cells to CT drugs. Hence, as already mentioned, it will be of great interest to address the importance of the control exerted by E4F1 on other genes, such as *FAAP100, SFR1* and *PARG,* in the DNA damage and replication stress responses. Future work in human cells will also have to integrate within the same study the relative importance of this transcriptional control and of all the other post-transcriptional regulations (for the moment mainly described in mouse models) [10,11] that E4F1 seems to exert on the DNA repair-process.

In the present state, all of the results suggest that through these multiple controls of the ATM/ATR–CHK1 pathway, E4F1 is a central actor in the survival of human cancer cells that heavily rely on this replication checkpoint machinery for survival. Indeed, the activation of the oncogenes combined with the inactivation of gate-keepers pathways (p53, pRB…) in these cells often results in elevated levels of replication stress and DNA damage. Hence, the targeting of the ATM/ATR–CHK signaling efficiently sensitizes the malignant cells, in particular the p53-deficient cells, to radiotherapy or chemotherapies, and is at the origin of numerous, promising clinical studies [24,25,26,27,28]. The central role played by E4F1 in the control of both the p53 and the ATM/ATR–CHK1 pathway suggests it could represent a key prognostic marker of the response to DNA-damaging agents and replication stress inducers and as such, a potential therapeutic target to enhance treatment efficacy. In this regard, it was recently reported that the *E4F1* gene is amplified in a significant proportion of human breast cancers that are wild type for *BRCA1/2* [11]. On the basis of our findings, it is clear that addressing the impact of this amplification in the ATM/ATR/CHK1-dependent response to treatment, is now worth exploring. 

## 3. Materials and Methods


**Cell culture and treatments**


The SUM159 Triple-Negative Breast Cancer (TNBC) cell line was obtained from Asterand Bioscience, UK, and grown in Ham’s F-12 medium (Gibco), supplemented with 5% fetal bovine serum, 10 μg/mL insulin, 1 μg/mL hydrocortisone, 100 μg/mL streptomycin and 100 units/mL penicillin and treated 24 h at the IC30 of Gemcitabine (140 nM, Sandoz), or Cisplatin (1.5 µM, Sigma Aldrich, St. Louis, MO, USA). The HCC38 TNBC cell line (ATCC-CRL-2314) was obtained from American Type Culture Collection and was grown in RPMI medium (Gibco), supplemented with 10% fetal bovine serum, 100 μg/mL streptomycin and 100 units/mL penicillin and treated 24 h with Gemcitabine (IC30, 180 nM; Sandoz), or Cisplatin (950 nM; Sigma Aldrich). When specified, the cells were treated with the lysosomal inhibitor E64 (10 µM, Sigma). All of the cell types were grown at 37 °C in a humidified atmosphere containing 5% CO_2_ and were regularly checked for the absence of mycoplasma contamination (MycoAlert^®^ Mycoplasma Detection Kit; Lonza).


**Knock-down and ectopic expression of E4F1**


The viral particles encoding pDONR221 wild-type E4F1-HA (the coding sequence of E4F1), or pLKO1 shRNAs either against human E4F1 (sh*E4F1*, Sigma-Aldrich mission clone TRCN013823 (targeted sequence in the 3′UTR of *E4F1* gene and sh*E4F1*#2, Sigma-Aldrich mission clone TRCN0000013826) or an irrelevant shRNA (sh*Ctrl*, clone shc002), were produced in HEK293T cell line, and added to exponentially growing SUM159 or HCC38 cells in presence of polybrene (8 µg/mL).


**Small Interfering RNA Transfection**


The SUM159 cells were transfected with 40 fmoles of small interfering RNA (siRNA) for human TTI2 SASI_Hs02_00323083 and SASI_Hs02_00323083_AS MISSION^®^ siRNA (Sigma Aldrich, St. Louis, MO, USA), or universal negative control #1 MISSION^®^ siRNA (Sigma Aldrich, St. Louis, MO, USA) as control. The transfection of the siRNA duplexes was performed for 5 h, using lipofectamine RNAiMAX transfection reagent (Invitrogen), according to the manufacturer’s instructions. The samples were collected 48 h post-transfection and probed for TTI2 protein level.


**Western Blots**


Western blots were carried out on the total cell extracts obtained by lysis on ice during 30 min (lysis buffer: Tris HCl pH7.4 50 mM, NaCl 100 mM, NaF 50 mM, β-glycerophosphate 40 mM, EDTA 5 mM, Triton X100 1%, Aprotinin 10 mg/mL, PMSF 100 mM, Leupeptin 1 mM, Pepstatin 1 mM), then spined and concentrations of the protein were evaluated by a BCA assay (kit BCA assay Thermo Scientific, Waltham, MA, USA). A total of 30 µg of protein samples were separated by SDS-page then transferred onto nitrocellulose membranes (Amersham) and incubated overnight at 4 °C with the primary antibody. The membranes were then washed and incubated with secondary antibody (Cell signaling, AntiMouse-HRP #70745; AntiRabbit-HRP #7076) for 2 h at room temperature and revealed by incubation with Chemiluminescent HRP Substrate (Millipore). The primary antibodies: Affinity-purified rabbit anti-E4F1 polyclonal antibody (Fajas et al., 2000); anti-CHK1 (Santa Cruz sc-8408); anti-P S345 CHK1 (Cell signaling #2348); anti-ATM (Cell signaling #2873); anti-ATR (Cell signaling #2790/ Genetex GTX128146); anti-TTI2 (Bethyl lab A303-476A); anti-GAPDH (Sigma-Aldrich G8795); anti-phospho-histone (Ser139) H2AX (Merck Millipore 05636); anti-PPP5C (Sigma-Aldrich HPA029065).


**Chromatin Immunoprecipitation (ChIP) assay**


Four sub-confluent 150 cm^2^ dishes of SUM159 cells (±4 × 10^7^ cells) wild type or knocked down for E4F1, were trypsinized and counted before being crosslinked for 6 min with 1% formaldehyde (SIGMA) directly added into the medium. About 750 mg of PDX tumors (B3804) were ground with an Ultra-Turrax^®^ (IKA) and crosslinked for 5 min in a PBS, 1% formaldehyde (SIGMA), 1% paraformaldehyde. The fixation was stopped with the addition of glycine at a final concentration of 125 mM for 5 min. The cells were rinsed twice with cold PBS. The cell nuclei were isolated by incubation for 5 min on ice, with 20 mM Hepes at pH 7.8, 10 mM KCl, 0.25% TritonX100, 1 mM EDTA, 0.5 mM EGTA and protease inhibitors. After centrifugation, the nuclei were re-suspended in 10 mM Tris at pH 8, 200 mM NaCl, 1 mM EDTA, 0.5 mM EGTA, and protease inhibitors, and incubated for 10 min on ice. The nuclei were lysed and the chromatin was extracted with a lysis buffer (10 mM Tris at pH 8, 140 mM NaCl, 0.1% SDS, 0.5% TritonX100, 0.05% NaDoc, 1 mM EDTA, 0.5 mM EGTA and protease inhibitors). The chromatin was sheared with a Vibra-Cell™ (bioblock) sonicator in 1.5 mL tubes floating in melting ice. A complete fragmentation of the genomic DNA (fragments below 600 base pair) was obtained after five series of 4 min pulses (15 s ON, 5 s OFF). An aliquot of sheared chromatin was decrosslinked and deproteinized for quality control before immunoprecipitation. The E4F1 and Mock (use for qPCR controls) ChIP are carried out within three tubes for each condition containing 4 μL of an affinity-purified rabbit anti-E4F1 polyclonal antibody incubated in the presence of 100µg ml of MEF chromatin and 45 μL of Dynabeads protein G. After overnight incubation, the immuno-precipitates were successively washed out with 1.5 mL of the five following buffers (W1: Tris at pH 8 10 mM, KCl 150 mM, NP40 0.5%, EDTA 1 mM; W2: Tris at pH 8 10 mM, NaCl 100 mM, NaDoc 0.1%, TritonX100 0.5%; W3a: Tris pH 8 10 mM, NaCl 400 mM, NaDoc 0.1%, TritonX100 0.5%; W3b: Tris at pH 8 10 mM, NaCl 500 mM, NaDoc 0.1%, TritonX100 0.5%; W4: Tris pH 8 10 mM, LiCl 250 mM, NaDoc 0.5%, NP40 0.5%, EDTA 1 mM; W5: Tris at pH 8 10 mM, EDTA 1 mM). The immunoprecipitated DNAs were eluted from the beads with 100 μL TE + 1% SDS. A total of 50 μL of input chromatin was diluted in 50 μL of TE + 2% SDS. The samples were decrosslinked overnight at 65 °C. They were diluted with 100 μL TE, 50 mM NaCl and 4 μg of RNAseA and incubated for 45 min at 37 °C and deproteinized with proteinase K (4 μg, 55 °C, 45 min). The proteins were removed with phenol–chloroform–isoamylic-alcohol and DNAs were recovered by chromatography with Nucleospin extract II columns (Macherey-Nagel). The DNA concentration was then determined with Qbit. For ChIP-qPCR experiments Real-time PCR was performed on a LightCycler 480 SW 1.5 apparatus (Roche) with SYBR^®^ Premix Ex TAK™ (TAKARA); 45 cycles of 95 °C for 4 s, 62 °C for 10 s and 72 °C for 30 s. For the ChIPseq experiments, 10 ng of immunoprecipitated DNA was sequenced on a HiSeq 2500 Illumina by Sequence By Synthesis at MGX platform (IGF, Montpellier).


**RT qPCR**


The total RNAs were isolated from the cells lysed in TRIZOL (Invitrogen). The cDNAs were synthesized from 1μg of total RNAs, using random hexamers and SuperScript III Reverse transcription (Invitrogen). The Real-time qPCRs were performed on a LightCycler 480 SW 1.5 apparatus (Roche) with SYBR^®^ Premix Ex TAK™ (TAKARA); 45 cycles of 95 °C for 4 s, 62 °C for 10 s, and 72 °C for 30 s. The results were quantified with a standard curve generated by serial dilutions of a reference cDNA preparation. Rpl13A transcripts were used for normalization. The fold change in gene expression was calculated as: Fold change = 2^−^^ΔΔ^^CT^. **Gene****Forward****Reverse***RPL13A*TTGATGGTCGAGGCCATCTCCTGCCAGAAATGTTGATGCCTTCACAGCG*E4F1*AAATCCGCTTCAGTGTGAGCAAGGAGGTGAATCACGGGTGAAGTCTCT*CHEK2*TGTTTCTGTTGGGACTGCTGGGTAACTTCTGCCCAGACTTCAGGAATG*CHEK1*GCTCCTCTAGCTCTGCTGCATAAAACTCTGACACACCACCTGAAGTGA*ATR*CCAGGCATCCTCCTATTTTTCTTTTCACCATGACGGTCTCC*ATM*CTGCAGAGAAACACGGAAACCCTGTGCACCATTCAAGAAC*TTI2*GTTAAGAGCGCCCTGCTACATCACAGCTTTGGGGAATTTT*PPP5C*AGCTCAAGACTCAGGCCAATGACTTGCCATAGTAGATGGCATTGCTGG


**RNA-sequencing**


The total RNAs were isolated from the cells lysed in TRIZOL (Invitrogen). A total of 2 µg of RNA was used to build the libraries, using the TruSeq Stranded mRNA Sample Preparation (Illumina) kit. The Fragment Analyzer (kit Standard Sensitivity NGS) and qPCR (ROCHE Light Cycler 480) were used to quantify fragments for libraries validation. The sequencing was performed on a HiSeq 2500 (Illumina) using the SBS (Sequence By Synthesis) technique. The following analyses were performed by HiSeq Control (HCS) and RTA (Illumina) Software. The control quality steps did not show any failures of contamination, quality of sequences and N base contents. For the statistical analysis of the differential genes expression, the R DESeq2 version 1.14.1 was used with a *p*-value of 5%.


**ChIP-seq Analysis**


The sequencing was performed at the MGX Montpellier GenomiX facility. The Fastq files (with 64 to 82 millions of reads) were first quality-checked with FASTQC (0.11.8). The reads were mapped to Hg38 Human genome with bowtie 2 (2.2.5) (bowtie2-t-q-p 6-fast), then sorted and indexed with samtools (1.32) and duplicated reads marked with picard MarkDuplicate. We used the HOMER suite (v4.10, 5-16-2018) for the following steps. Peak calling was performed with findPeaks (-style factor), bed files were generated with pos2bed.pl and annotation was realized with annotatePeaks.pl. Motif finding was performed with findMotifsGenome.pl. Bigwigs were generated with Deeptools bamCoverage and the peaks around the TSS plot were generated with ChIPSeeker (1.28.3) R package.


**Apoptosis assay**


Three days after a lentiviral infection of a shRNA targeting E4F1 or a control, the shRNA cells were exposed to the corresponding drugs for 24 h then grown in a drug-free medium for 24 h. The cells were washed with PBS, trypsinized, pooled with floating cells and counted. The annexin V and propidium iodide dual labelling of apoptotic cells was conducted using the annexin V-FLUOS Staining Kit from Roche Applied Science according to the manufacturer’s instructions. Two million cells were washed twice with PBS and once with binding buffer (10 mM HEPES, 140 mM NaCl, 5 mM CaCl_2_, pH 7.4). They were then incubated at room temperature in the dark for 30 min in 2 μL annexin-V-Fluos reagent and 2 μg/mL propidium iodide in binding buffer. Analysis of 10,000 events was performed on a Gallios (Beckman Coulter) flow cytometer. DNA fluorescence was collected in logarithmic mode and cell viability was quantitated using Kaluza software (Beckman Coulter, Brea, CA, USA).


**Cell cycle assay**


Three days after a lentiviral infection of a shRNA targeting E4F1 or a control shRNA cells were exposed to the corresponding drugs for 24 h. Cells were washed once with ice-cold PBS, trypsinized and counted. One million cells per sample were fixed in ice-cold 70% ethanol and stored at −20 °C, and were then washed in PBS before being suspended in 0.5 mL staining solution (4 μg/mL propidium iodide, 200 μg/mL RNAse A in PBS) and incubated at 37 °C for 30 min. The analysis of the 10,000 events was performed on a Gallios (Beckman Coulter) flow cytometer. The DNA fluorescence was collected in linear mode, using a doublet discrimination gate and cell-cycle distribution, was analyzed using Kaluza software (Beckman Coulter).


**Immunofluorescence analyses**


The exponentially growing SUM159 cells were seeded on 10 mm coverslips in 6-well plates, exposed to the corresponding drugs for 24 h. The cells were washed twice with PBS, incubated in 3% paraformaldehyde in PBS for 15 min, washed in PBS for 5 min, blocked with PBS containing 3% bovine serum albumin and 0.5% triton X-100, and incubated 1 h at room temperature with anti-phospho-histone (Ser139) H2AX (Merck Millipore; 05636) antibody, anti-ATR (Genetex; GTX128146) antibody or anti-53BP1 (Novus; NB 100-904). The slides were washed, incubated with a FITC AlexaFluor 488-conjugated goat anti-rabbit or anti-mouse IgG antibody for 1 h at room temperature, and washed in PBS. The slides were mounted in Mowiol and image acquisition was performed on an ApoTome microscope (Zeiss, Jena, Germany), and the data were collected and analyzed with Zen software (Zeiss).


**In vivo study, PDX model**


All of the experimental procedures were conducted in compliance with recommendations issued by the *Comite d’Ethique Regional Languedoc-Roussillon*—*Comite d’Ethique pour l’Experimentation Animale* (CEEA-036) and the Guidelines for the Welfare of Animals in Experimental Neoplasia. Female Swiss nu/nu mice (Charles River Laboratories, Wilmington, MA, USA) were housed under clean room conditions in sterile, individually ventilated cages. The animals received sterile irradiated chow and water ad libitum. 

The tumors were passaged onto a further cohort of mice before the graft volume reached 2000 mm^3^ [44]. A fragment (~8 mm^3^) of B3804 triple-negative breast cancer PDX tumor [44,48] was implanted into the inter-scapular fat pads of 3–4-week-old female SwissNude mice. When the tumors reached approximately 150 mm^3^ the animals were randomly assigned into two groups of three mice. The mice were then administered once with vehicle or Gemcitabine at a dose of 60 mg/kg. The animals were euthanized 48 h after the administration injections and tumors were instantly frozen.


**Statistical analysis**


Data were expressed as mean values ± SD of three independent experiments and were analyzed with GraphPad Prism (Dotmatics) using one way ANOVA test for annexinV, cell cycle and RTqPCR analysis and using a two-tailed unpaired Student’s *t*-test for immunofluorescence staining analysis.

## Figures and Tables

**Figure 1 ijms-23-09217-f001:**
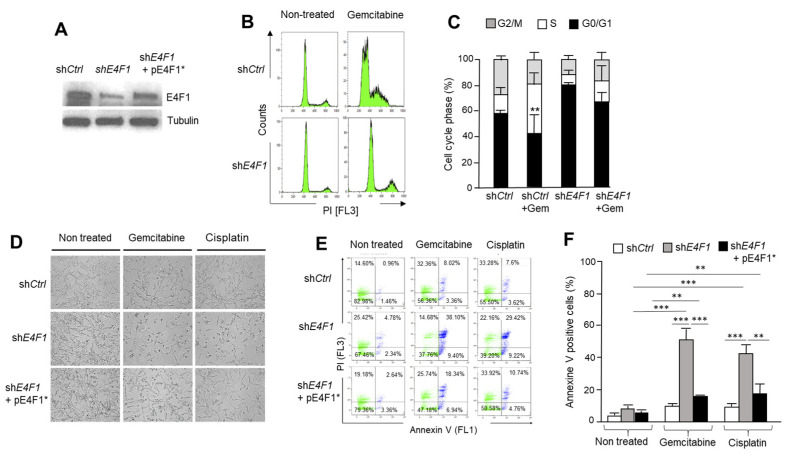
Depletion of the transcription factor E4F1 sensitizes SUM159 Human Triple-Negative Breast Cancer (TNBC) cell line to the chemotherapy drugs Gemcitabine and Cisplatin. (**A**)—Depletion of E4F1 in SUM159. Immunoblot analysis of E4F1 protein level in protein extracts prepared from SUM159 cells treated with either a control shRNA (sh*Ctrl*) or a shRNA directed against E4F1 (sh*E4F1*), or with a combination of the latter sh*E4F1* with an expression vector encoding an E4F1 cDNA resistant to this sh*E4F1* (pE4F1*). Extracts were prepared 4 days after cell infection with lentiviral vectors expressing shRNAs. shRNA directed against human E4F1 have been previously validated [18]; (**B**,**C**)—E4F1-depleted SUM159 cells fail to arrest in S-Phase upon Gemcitabine treatment. Three days after sh*Ctrl* or sh*E4F1* treatment, cells were exposed to sub-lethal dose of Gemcitabine (140 nM, IC30) for 24 h; (**B**) Representative flow cytometry analysis (PI staining) of cell cycle distribution in populations of cells collected at 24 h after start of Gemcitabine treatment. Of note FACScan gating settings were meant to exclude sub-G1 cells; (**C**) Quantitation of the distribution of cells in G2/M, S and G0/G1 cell cycle phases was performed with the FlowJo software. Results are expressed as mean values ± SD of three independent experiments. ** *p* < 0.01, by a one way ANOVA test performed on GraphPad Prism; (**D**–**F)**—shRNA-mediated depletion of E4F1 sensitizes TNBC SUM159 cells to cytotoxic drugs, an effect rescued by the co-expression of the sh*E4F1*-resistant pE4F1* construct (see also Appendix A on another TNBC cell line, HCC38). Three days after sh*Ctrl*, sh*E4F1* or sh*E4F1* + pE4F1* transduction, cells were exposed to sub-lethal doses of Gemcitabine (140 nM, IC30) or Cisplatin (1.5 mM) for 24 h and assessed for cell death by (**D**) Phase contrast microscopy, (**E**,**F**) Flow cytometry analysis for apoptosis /necrosis by staining dying cells with annexin V/7-AAD and propidium Iodide (PI). (**E**) Representative flow cytometry profile of annexin V/7-AAD/PI –positive cells, and (**F**) quantitation (FlowJo software) of three independent experiments expressed as mean values ± SD of three independent experiments. ** *p* < 0.01, *** *p* < 0.001 as determined by a one way ANOVA test performed on GraphPad Prism.

## Data Availability

Data were posted in the database Gene Expression Omnibus under the number GSE128159 (superserie) comprising GSE128099 (RNA-seq of sh*E4F1* or sh*Ctrl* SUM159 triple-negative breast cancer cell line treated or not with Gemcitabine) and GSE128104 (ChIP-seq of E4F1 in SUM159). Transcriptomic and ChIP-seq analyses of E4F1 target genes in primary mouse embryonic fibroblasts (MEFs) and in p53^−/−^; Ha-RasV12-transformed MEFs, are available as GSE57228, as previously published in Refs. [17,18].

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
