# Peer review of "Multi-Level Control of the ATM/ATR-CHK1 Axis by the Transcription Factor E4F1 in Triple-Negative Breast Cancer"

_ijms, 2022, doi:10.3390/ijms23169217_

Round 1
Reviewer 1 Report
The findings appear to be interesting. Specific points that the authors need to address are as follows:
1. Most of the experiments have been done in SUM159 cell line. Additional TNBC cell line should be used to validate the key findings of the study.
2. How E4F1 regulates the phosphorylation of γH2AX and CHK1 should be investigated in detail.
3. Whether depletion of the transcription factor E4F1 can sensitize SUM159 to doxorubicin and docetaxel should also be analyzed?
4. The limitations associated with the study should be discussed.
5. Proper statistical analysis should be conducted for all the figures.
6. Typographical errors were found throughout the manuscript and should be corrected.
Reviewer 2 Report
The manuscript sent for peer-review is concerned around the E4F1 transcription factor and its controlled program of ATM/ATR – CHK1 signaling pathway and DNA damage-stress response in triple negative breast cancer cells, which was evaluated by ChIP-Seq and RNA-Seq methods.
This is a difficult paper to read - the paper could be abbreviated considerably and the results presented much more clearly.
I understand that the work on the project takes many years, but the authors cite too many of their own works (over 10?) – namely 8, 9, 12, 13, 14, 15, 17, 18, 19, 21, 22.
The introduction, apart from a few sentences, consists mainly of summaries of their previous activities on this topic, and does not contain much objective information.
The numbering of the cited works is not consecutive, e.g. line 46 and further: … and tissue homeostasis (13,18)… unless there should be a dash instead of a comma.
Lack of clear hypothesis in Aims – what was your primary expectation about influence of disrupted ATM/ATR-CHK1 cascade in breast cancer cells? Could authors provide such information?
Line 441 – please provide a name of the mycoplasma detection kit
Do the cell lines have a certificate of authenticity number of ECACC or other? As the main points and conclusions from this study are based on cell lines investigations, this information is crucial.
RT qPCR was performed using SYBR green – have authors checked the specificity of obtained products by melting curves analysis? The reaction melt curves should be shown to rule out any differences in specificity or non- specificity of performed reactions. This can be added as supplementary data, but is important. How was the gene expression calculated?
Why the cells were exposed to drugs for 24 hours, why they were not preliminary checked after 12h? Please justify this.
Please unify the naming – QPCR to qPCR – through manuscript
Please describe exactly where did the mice come from, where were they bought?
In my opinion in the materials and methods section there should be a separate subchapter related to statistical analysis – please provide such description.
Despite all the above-mentioned issues, overall this is a well-designed and performed research, which can be of interest to readers, but needs to be improved before being published.
Round 2
Reviewer 2 Report
Thank you for the concise and full responses to my comments. Although I still find the large amount of self-citations disturbing.
Nonetheless, the changes are sufficient to accept your manuscript.
Some minor issues:
ATR-CHK1 abbreviation should be explained in the first mention (line 73), not in line 79.
TNBC abbreviation is already explained in the line 76, there is no need to repeat it (line 102,110, 284, 533)
Also, please be consistent: TNBC as Triple Negative Breast Cancer or triple-negative breast cancer.
Chemotherapy as CT is also explained in line 108, no need to repeat it (line 113, )
The table between lines 623-624 is missing a title and a reference in the manuscript text.
Line 634 – I believe that p-value should not be given as a percentage